# Self-selective formation of ordered 1D and 2D GaBi structures on wurtzite GaAs nanowire surfaces

Yi Liu [1], Johan V. Knutsson[1], Nathaniel Wilson[2], Elliot Young[2], Sebastian Lehmann [1], Kimberly A. Dick[3], Chris J. Palmstrøm[2,4], Anders Mikkelsen[1] & Rainer Timm [1✉]

Scaling down material synthesis to crystalline structures only few atoms in size and precisely positioned in device configurations remains highly challenging, but is crucial for new applications e.g., in quantum computing. We propose to use the sidewall facets of larger III–V semiconductor nanowires (NWs), with controllable axial stacking of different crystal phases, as templates for site-selective growth of ordered few atoms 1D and 2D structures. We demonstrate this concept of self-selective growth by Bi deposition and incorporation into the surfaces of GaAs NWs to form GaBi structures. Using low temperature scanning tunneling microscopy (STM), we observe the crystal structure dependent self-selective growth process, where ordered 1D GaBi atomic chains and 2D islands are alloyed into surfaces of the wurtzite (Wz) {11$\bar{2}$0} crystal facets. The formation and lateral extension of these surface structures are controlled by the crystal structure and surface morphology uniquely found in NWs. This allows versatile high precision design of structures with predicted novel topological nature, by using the ability of NW heterostructure variations over orders of magnitude in dimensions with atomic-scale precision as well as controllably positioning in larger device structures.

[1] Department of Physics and NanoLund, Lund University, Lund, Sweden. [2] Materials Department, University of California-Santa Barbara, Santa Barbara, CA, USA. [3] Centre for Analysis and Synthesis and NanoLund, Lund University, Lund, Sweden. [4] Department of Electrical and Computer Engineering, University of California-Santa Barbara, Santa Barbara, CA, USA. ✉email: rainer.timm@sljus.lu.se

Site-selected formation of semiconductor nanostructures has so far mainly been realized by two types of approaches: Nanostructures with full control over the spatial position of each atom have been produced by means of atom manipulation[1], where individual atoms are placed or re-positioned on a clean semiconductor surface using an STM probe tip. In this way, e.g., single atom transistors based on P doping atoms in Si have been demonstrated[2,3], and quantum dots (QDs) formed of chains of In atoms on an InAs surface have been studied[4]. While these approaches give fascinating insight into quantum processes, they are not suitable for large-scale industrial device processing, and demanding to implement even for scientific purposes. Scalable approaches, on the other hand, are typically based on complex processing recipes using lithography and etching steps[5]. Many impressive examples can be found, including InAs nanoribbons based on multi-step epitaxial lift-off and overgrowth demonstrating high-performance transistor behavior[6], or self-assembled QDs aligned in chains by substrate strain engineering[7] or cleaved-edge overgrowth[8], aiming for optoelectronic application. In these approaches, however, the lateral positioning is limited by the spatial resolution of the lithographic processes, and the nanostructures formed on top of the substrates are at least several nm high and contain millions of atoms, thus being far from atomic-scale precision, which has to be the ultimate goal for quantum devices[9]. 2D small islands and 1D chains of atoms have been grown on macroscopically homogeneous group IV semiconductor surfaces[10–12]. However, formation of such nanostructures with precise spatial control on III–V or other compound semiconductor heterostructures remains an open challenge.

In an alternative approach, we use side facets of crystal phase engineered III–V semiconductor NWs[13] as templates for site-selected overgrowth, after mechanical transfer of the NWs from the growth sample onto a suitable substrate[14]. Such NW heterostructures can be grown with atomically sharp transitions between cubic zincblende (Zb) and hexagonal Wz segments and tunable length of the corresponding segments, providing the desired spatial precision and control. The wires themselves can be placed intentionally in devices after or even during growth[15]. Interestingly, the various surface facets of both Zb and Wz crystal phase[13,16–18] of III–V semiconductor NWs offer an extra degree of freedom for surface-related growth phenomena. Previous STM studies have successfully resolved the atomic structure of these facets and the Wz/Zb interface on GaAs[14,19] and other III–V[16,17,20] NWs, showing that the Zb {110}, Wz {11$\bar{2}$0}, and Wz {10$\bar{1}$0} facets are non-polar and usually appear unreconstructed. Step density and surface morphology can also be controlled. However, previous studies have not found any significant ordered incorporation in the atomic lattice[18].

Among the various possible material combinations of the III–V semiconductor toolbox, alloys with a high bismuth content have recently gained much attention, mainly due to their large spin-orbit splitting and resulting material properties. GaBi compounds, for example, have been predicted to show band inversion and topological behavior[21,22]. However, III–V alloys with a high Bi content (>20% of the group-V atoms) could not yet be realized[23,24], and especially GaBi has been considered as thermodynamically unstable[25–27], while Bi is also known to act as surfactant in the GaAs/InAs material system[28]. Alternatively, one can consider to deposit Bi atoms onto the surfaces of binary III–V compounds and thus obtain a high Bi concentration at the surface. So far, Bi deposition on the non-polar GaAs(110) surface was reported to result in the growth of terraces and islands of pure Bi[29], while after Bi deposition on polar surfaces such as GaAs(001) or {111} mainly Bi-terminated surface reconstructions were observed (see also Supplementary Note 1).

Here, we report the study of Bi incorporation in the surface of GaAs NWs where we find ordered 1D GaBi chains and 2D GaBi islands, which form self-selectively on the Wz {11$\bar{2}$0}-type NW facets, in contrast to randomly distributed Bi atoms in the Zb {110} facets, which are found also in bulk. The present study focuses on the materials aspects of position controlled growth of few atom small nanostructures, but the surface nanostructures observed here are also highly promising for application in quantum information, since they provide local areas of pure GaBi. Still, the main point is how their self-selective formation only on Wz segments, in combination with the tunability of the structure of the NW template, opens the path towards monolayer thin nanostructures with variable width, atomically sharp borders, and promise for exotic electronic phases of matter.

## Results

**Bi atoms incorporation into GaAs NW surface.** GaAs NWs (see Fig. 1a and Supplementary Fig. 1) were grown by metal-organic vapor phase epitaxy (MOVPE) via Au particle-assisted growth[30]. The NWs consist of a long Wz segment followed by a Zb segment including smaller Wz insertions as well, as seen in Fig. 1a. The NWs have been broken off and transferred onto GaAs{111}B substrates, exposing their side facets to the STM tip as sketched in Fig. 1b. Surface morphology has been studied with atomic resolution using STM at 5 K in ultrahigh vacuum (UHV). Prior to the studies, the native oxide was removed from the NW surfaces by annealing in atomic hydrogen[18]. In order to investigate Bi incorporation into the GaAs surface, we deposited elemental Bi from a thermal evaporator onto NW samples annealed to 250°C. We found that the majority of NWs had the {11$\bar{2}$0}- and {110}-type facets exposed in Wz and Zb segments, respectively, while the more seldom {10$\bar{1}$0}-type facets were observed at a 30-degree angle inclined to normal. An overview and a zoom-in STM image of a {11$\bar{2}$0}/{110}-type facet with an atomically sharp Wz/Zb interface are shown in Fig. 1c, d, respectively. Terraces with monolayer high surface steps were found on all facets, showing unreconstructed morphology which has the same patterns as expected from oxide-free pure GaAs NWs[14], indicating no effect from Bi incorporation on the general surface morphology (see Supplementary Fig. 2).

Bi atoms surface incorporation can be found on both Zb and Wz facets, as shown in Fig. 1c, d. These STM images are obtained at negative sample bias which corresponds to filled state imaging, revealing the group-V atoms in the top most surface layer[31]. Importantly, a number of bright sites can be seen within the rows of As atoms on the surface after Bi deposition. Such bright protrusions are consistent with the presence of an adatom or different atomic species incorporated within the atomic lattice[32], and we associate them with individual Bi atoms. These Bi atoms protrude the surrounding GaAs surface by only about 40–50 pm, as shown by the line profiles of Fig. 1e for {110}- and {11$\bar{2}$0}-type facets as well as Supplementary Fig. 3 for {10$\bar{1}$0}-type facets, independent of tunneling voltage and current settings. Thus, we can exclude the influence from the variation of local density of states (LDOS), and conclude the protrusion indeed comes from tomographic changing. Considering the well-studied group-V-exchange reaction, we interpret the brighter protrusions in the As lattice as Bi atoms that have entered the surface via a Bi-for-As exchange in the top most surface layer, forming local Ga-Bi bonds. The brighter contrast of the Bi atoms compared to the surrounding As atoms originates from the larger atomic radius of Bi. Other similar group-V exchange reactions have been observed before, such as Bi incorporation in As sites in diluted GaAsBi[33] and Sb in GaAs[18]. Bi adatoms on top of the surface, in contrast, would result in a much larger height. Bi atoms situated below

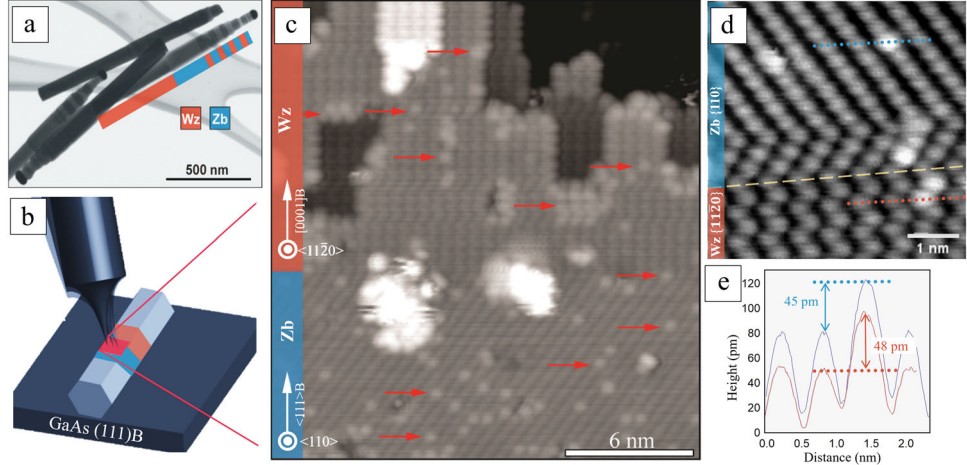

**Fig. 1 NW sample preparation and NW surfaces. a** Scanning electron microscopy image in transmission mode with crystal phase contrast of GaAs NWs, after transferring onto a suitable sample holder. Brighter and darker contrast corresponds to Zb and Wz, respectively, as confirmed previously by transmission electron microscopy. **b** Schematic description of the STM imaging setup showing the STM tip accessing a NW side facet. Orange (blue) color indicates Wz (Zb) segment. The red square demonstrates the area being scanned by STM. **c** Atomic-scale STM image of a Bi-exposed GaAs NW with a Wz {11$\bar{2}$0}- (top) and Zb {110}- (bottom) type facet, with some Bi atom sites marked by red arrows. Monolayer-high surface steps are found on the {11$\bar{2}$0}-facet, resulting in terraces of four different heights. Two small bright patches of remaining oxide or adsorbates can be seen close to the Wz–Zb interface. **d** A zoom-in filled state STM image of the Wz/Zb interface, showing the group-V atoms. The yellow dashed line indicates the interface between Wz {11$\bar{2}$0}- (bottom) and Zb {110}- (top) type facet. The brighter dot-like protrusions are Bi incorporation sites, which match well with the As lattice sites. The gray scale range, i.e., the brightest pixel to darkest pixel for (**c**) is 670 pm, and for (d) is 104 pm. (e) Line scan along the blue (orange) line in (d) showing the height of the protruded atom in Zb (Wz) segment. STM images were obtained at (**c**) $V_T = -3.5$ V, $I_T = 100$ pA and (**d**) $V_T = -4.4$ V, $I_T = 75$ pA.

the surface layer would give rise to a much smaller outward relaxation (towards the vacuum), which should still be detectable with STM down to at least 3 bilayers[34], however, we find no such protrusions. These results suggest Bi-incorporation on GaAs NWs after growth as a successful approach for achieving local GaBi formation, which has been considered unstable in bulk[25–27], confined within the top most atomic layer.

**Distinct differences in the incorporation of Bi atoms between the Wz {11$\bar{2}$0}- and Zb {110}-type facets.** On the {110}-type facets, Bi atoms are found in a random distribution in the middle of the terraces, as confirmed by autocorrelation analysis (see Supplementary Fig. 4), with some additional Bi agglomeration at step edges (see Fig. 2a). In contrast, very few single Bi sites are observed on the {11$\bar{2}$0}-type facets, instead a short-range ordering with neighboring sites in all directions is found to be preferable, which results in small 1D chains or 2D islands of local GaBi structures made up with a few tens of atoms (see Fig. 2d); only about 4% of the Bi atoms were found to sit individually in the {11$\bar{2}$0}-type surfaces (see Supplementary Fig. 5 and Supplementary Table 1). The atomic lines and zig-zag pattern in Fig. 2a and Fig. 2d are due to the specific side facets exposed in NWs, the corresponding crystal structure models can be seen in Figs. 2b and 2e, respectively. It appears favorable for the incorporated Bi atoms to be positioned next to each other in the nearest neighbor sites on the Wz {11$\bar{2}$0} facet, while Bi prefers to stay at single sites on Zb {110} facets. Statistical results were obtained by analyzing STM images from large areas of the NW facets across five different NWs (see Supplementary Fig. 6 and Supplementary Table 2). It is found that Bi in average replaces 6%, 6%, and 1% of the As atoms on {11$\bar{2}$0}-, {110}-, and {10$\bar{1}$0}-type facets, respectively. This is in all cases significantly less than half a monolayer, which was the nominal amount of deposited Bi. This suggests that only a small fraction of the deposited Bi atoms will overcome the energy barrier for the group-V exchange reaction and get incorporated into the surface. An example of a single atomic vacancy in the middle of a flat {11$\bar{2}$0} terrace, which has triggered

formation of a 2D GaBi island directly beside it, can be seen in the inset of Fig. 2d. Aggregation of Bi atoms can also be found surrounding vacancies in the {110} terraces, as marked with a red arrow in Fig. 2a. This indicates a much lower exchange energy barrier at the step edges and atomic vacancies as compared to that in a continuous surface, enhancing Bi-for-As exchange. This is due to the generally higher surface potential of As atoms on the higher terrace near the step edge[35] and around vacancies. Furthermore, Bi atoms will have more freedom for strain relaxation when incorporating on the step edges.

**The mechanism for the different Bi incorporation on the Wz and Zb facets.** The behavior can be explained by a model in which Bi, after landing on the surface, diffuses around, upon which it can either substitute with As atoms in the GaAs lattice or evaporate from the surface again. The lower than nominally deposited Bi surface content would support this picture, as not all the Bi incorporates into the surface. For the Wz{11$\bar{2}$0}-type facets a much higher density of Bi atoms is found on all the step edges facing [0001]A/B directions compared to other step edges (see Fig. 2d). Further STM measurements, performed on Wz {11$\bar{2}$0}-type NW facets after Bi deposition resulting in a coverage of less than 10% (nominally), can be seen in Supplementary Fig. 8. They confirm that the initial sites for Bi incorporation are step edges facing [0001]A/B directions. More specifically, sites on step edges terminated with As atoms (i.e., [0001]B) are more preferable compared to edges terminated with Ga atoms (i.e., [0001]A). This indicates that the energy barrier is low for Bi-for-As exchange at edges facing [0001]B. 2D GaBi islands or 1D GaBi chains are found adjacent to such step edges, where Bi atoms are occupying neighboring group-V lattice sites along the [0001]A direction. This cannot be explained by very short range energetics, as the nearest neighbor configuration is identical for the Wz{11$\bar{2}$0} and Zb{110} surface, while ordered Bi agglomeration is only observed on the Wz facet. Instead, more long range energetics have to be responsible for the markedly different behavior on the two surfaces. Longer range elastic strain energy

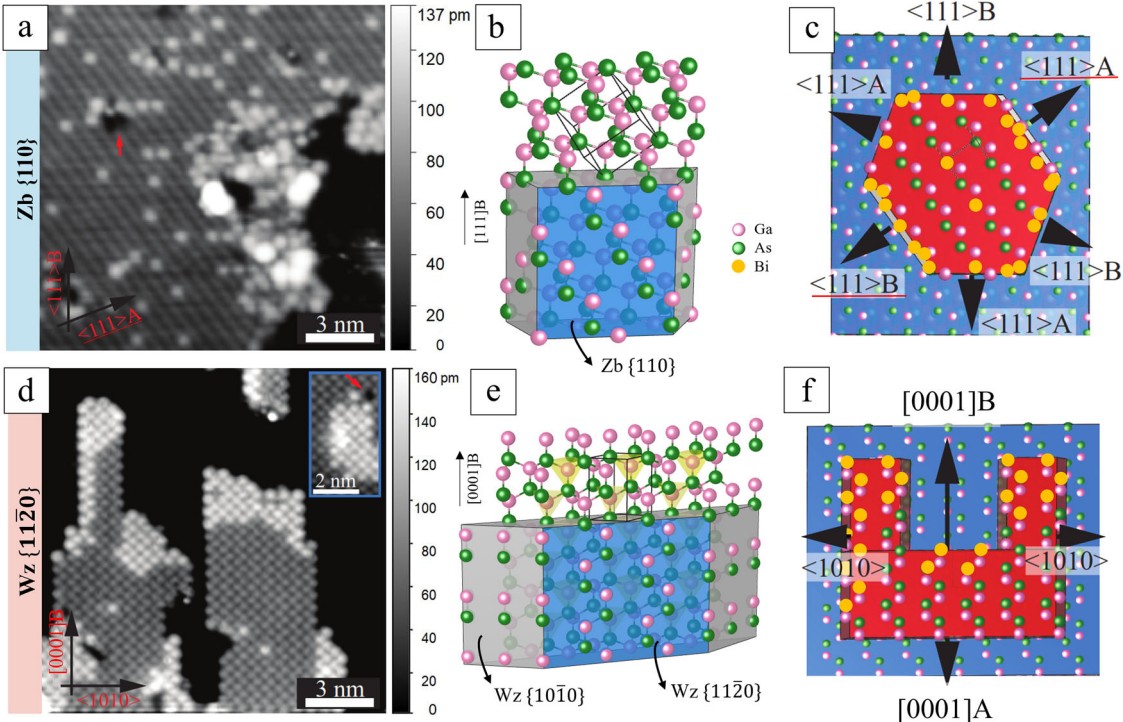

**Fig. 2 Bi incorporates into the surface layer of GaAs NWs. a, d** shows STM images of the Zb {110}-type surface and of the {11$\bar{2}$0}-type surface, respectively, including surface steps along different directions between the top terrace and a one monolayer lower terrace. The lower terraces appear black here due to contrast enhancement, the height scale bars are shown on the right side. Different contrast settings of the same STM image can be found for comparison in Supplementary Fig. 6. The many small bright protrusions correspond to Bi atoms. Red arrows in (**a**) and (**d**) point at atomic vacancies. The inset in (**d**) indicates a 2D GaBi island near a vacancy in the middle of a flat terrace. **b, e** illustrate the GaAs 3D crystal structure for (**b**) Zb and (**e**) Wz phase, overlaid with the NW shape. **c, f** show 2D models corresponding to the surfaces seen in (**a**) and (**d**). The red and blue areas represent the top terrace and the second atomic layer, respectively, of the NW surface, separated by surface steps. An arbitrary shape is chosen for the top terrace, but the possible directions of the surface steps are determined by the crystal structure and correspond to those visible in (**a**) and (**d**). Incorporation of Bi atoms (yellow dots) is illustrated, occurring at vacancies and via atomic step edges facing ⟨111⟩A/B directions on the Zb{110}-type surface (**c**) and step edges facing the [0001] B growth direction on the Wz {11$\bar{2}$0}-type surface (**f**), as motivated by the positions of the Bi atoms observed in (**a**) and (**d**). The direction indexes with red underline in (**a**) and (**c**) present the projections of <111>-type directions in {110} plane. In (**b**), (**c**), (**e**), and (**f**), pink (green/yellow) spheres depict Ga (As/Bi) atoms. $V_T = -4.4$ V, $I_T = 100$ pA for all STM images.

minimization as a mechanism is not a reasonable explanation either, as the incorporation of several Bi atoms in close vicinity would lead to a larger overall strain and a broader distribution of Bi near the edges would be preferred, in strong contrast to the almost perfect ordering observed. Here it can further be noted that the lattice difference between GaBi and GaAs is larger than for many other more commonly studied systems such as Si/Ge and InAs/GaAs. Other possibilities are pure geometric effects[36] due to limited diffusion on the surface in some directions, but this is also difficult to reconcile with the highly ordered islands observed. Instead, an intermediate situation must be considered in which the zig-zag motif of the top layer of the Wz{11$\bar{2}$0} surface (see Fig. 2f) must lead to a weakening of the Ga–As bonds of the As atoms placed next to the recently incorporated Bi atoms, thus the barrier for incorporation in an adjacent site would be lowered. In contrast, the rows on the Zb{110} surface have a different symmetry (when considering the formation of a cluster of three Bi atoms or more), which then cannot be energetically favorable. Interestingly, the pronounced effect of next nearest neighbor interaction is fundamental to crystal formation, it is the basic distinguishing factor between Wz and Zb crystal structures.

Bi can also exchange As surface atoms in the middle of a {11-20} terrace, however, this process has a much lower probability and will only occur for large terraces and depend on temperature, as it is competing with evaporation and edge integration (as seen in Supplementary Fig. 5). The model would then also explain why

agglomeration does not start to occur in the presence of a single Bi atom in the middle of the surface. Morphologically, monolayer high terraces are generally favored elongated along the [0001]A/B directions on the naturally formed Wz{11-20} facets on GaAs and InAs NWs[14,17,18]. 2D islands result from Bi incorporation at all group-V atom positions along a step edge facing [0001]B with subsequent lowering of energetics for Bi incorporation in adjacent sites. If a Bi atom is incorporated at a corner site, it appears especially favorable for another Bi atom to substitute As along a step edge facing a ⟨10$\bar{1}$0⟩ direction, and a 1D chain is formed along the edge. 1D chains might energetically be favorable over 2D islands due to strain relaxation at the edge. Vacancies in principle trigger Bi incorporation in the same way, as they can generate step edges facing different directions (as seen in the inset of Fig. 2d). On the {110}-type facets, Bi-for-As exchange occurs at all step edges, but adjacent group-V lattice sites are not preferred for subsequent Bi incorporation. Instead adsorption in random sites on the terrace becomes relatively more likely, as well as Bi-Bi clustering, where Bi atoms are incorporated outside group-V lattice sites. This is in agreement with the lack of large ordered GaBi structures or GaAsBi alloys with high Bi content reported previously for the standard Zb surfaces[37]. While this qualitative model explains the observed results, it also indicates that future detailed calculations of the surface energetics would be highly interesting and could be used to tailor the behavior.

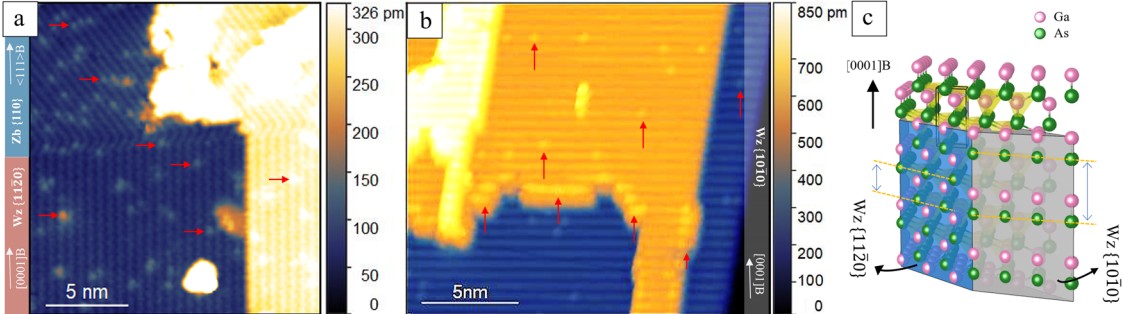

**Fig. 3 Atomically resolved STM images of Bi incorporation in GaAs NW surfaces with a lower step density. a** shows the interface between a Wz {11$\bar{2}$0}-
(bottom) and a Zb {110}- (top) type facet. In this image, no step edge facing [0001]A/B directions is present on the Wz {11$\bar{2}$0} facet. Almost all Bi atoms
are incorporated at individual sites. In contrast, the Zb {110} facet includes more Bi atoms, in the form of unordered Bi clusters along the step edges or
random individual Bi incorporation sites. **b** shows the surface of a Wz {10$\bar{1}$0}-type facet. At all step edges facing [0001]-type directions, most of the As
atoms are substituted by Bi, while very few Bi incorporation sites are found on the other step edges or within the terraces. The slightly tilted appearance of
the step edges is explained in Supplementary Note 2. Some Bi atoms are indicated in (**a**) and (**b**) by red arrows. The height scale bars are shown on the
right side in both (**a**) and (**b**). **c** sketches the 30° edge between a Wz {10$\bar{1}$0}- and a {11$\bar{2}$0}- type facet, pink (green) spheres depict Ga (As) atoms. Every
layer of Ga and As (or Bi) atoms along the [0001]B growth direction can be seen on the {11$\bar{2}$0} surface, resulting in the zig-zag chain pattern visible at high
resolution as e.g., in Fig. 2d due to the AB-stacking sequence of the atomic layers. However, only every second layer of Ga and As (or Bi) atoms can be seen
in the top most plane of the {10$\bar{1}$0} surface. STM images were obtained at (**a**) $V_T = -4.4$ V, $I_T = 50$ pA and (**b**) $V_T = -4.5$ V, $I_T = 90$ pA.

**Bi incorporation on GaAs NW surface with lower step density**.
To further investigate the influence of surface orientation and
step density on the Bi incorporation, we studied another type of
GaAs NWs with larger surface terraces and a lower step density.
An STM image of an interface of {11$\bar{2}$0}/{110}-type facets on such
a GaAs NW is shown in Fig. 3a, which presents a similar dis-
tribution of Bi atoms on the Zb {110} facet as before (Figs. 1c and
2a), but a significantly different configuration for the Wz {11$\bar{2}$0}
facet (compared with Figs. 1C and 2C): Here, incorporated Bi
atoms are only found at individual sites, which appear to be
randomly distributed. The only step edges, which are facing the
<10$\bar{1}$0>-type directions, show the same low density of Bi sites as
the flat terraces, without any short-range ordering of Bi atoms.
Furthermore, here in average 7% of the surface As atoms were
replaced by Bi on the {110}-type facets, but only 3% on the
{11$\bar{2}$0}-type facets, while the ratio was the same for both types of
facets for the NWs with higher step densities (see Supplementary
Fig. 7 and Supplementary Table 3). These results confirm that the
step edges facing the [0001]A/B directions on Wz {11$\bar{2}$0} facet,
which are missing here due to the larger terrace size, play a
critical role on triggering Bi incorporation and forming ordered
GaBi structures.

Yet another behavior of Bi incorporation is observed on Wz
{10$\bar{1}$0}-type facets, as shown in Fig. 3b. Although Bi atoms
successfully incorporate into the first sites on the step edges
facing [0001]A/B directions, no ordered GaBi structures
extending further into the terraces can be found. We attribute
this different Bi incorporation to the different surface structure
of Wz {11$\bar{2}$0} and Wz{10$\bar{1}$0} facets, as shown in the model of
Fig. 3c. On the Wz{10$\bar{1}$0} surface, the horizontal As (or Ga)
atomic chains are separated along the [0001]A/B directions by
0.55 nm, which is twice the distance between neighboring As (or
Ga) atoms along the [0001]A/B direction in the zig-zag motif on
the Wz{11$\bar{2}$0} surface. The much larger gap results in a smaller
nearest neighbor effect of Bi atoms along [0001]A/B directions
on the {10$\bar{1}$0} facet. This is consistent with the Bi incorporation
mechanism discussed above.

Finally, we want to further explore the conditions under which
Bi incorporation into the sample surface can occur at all. We
found that if the sample is at too low temperature (e.g., room
temperature) during Bi deposition, only metallic Bi clusters will
be formed on the surface. On the contrary, if the sample

temperature is too high (e.g., 400 °C), Bi atoms tend to evaporate
directly when landing on the GaAs surface, thus resulting in a
very low Bi deposition and incorporation rate. Synchrotron based
nano-focused X-ray photoelectron spectroscopy (XPS) has been
performed to study the Bi incorporation process as well, as shown
in Supplementary Fig. 9. The presence of Bi atoms in the NW
surface after Bi deposition was confirmed from Bi 4 f core level
spectra. For these experiments, GaAs NWs have been transferred
onto Si substrates prior to Bi deposition. Interestingly, 2D maps
obtained by scanning photoelectron microscopy (SPEM), shown
in Supplementary Fig. 9, demonstrate a much higher Bi signal at
the GaAs NW as compared with the surrounding Si substrate.
This can be explained by the Bi incorporation process on NWs,
such that the incorporated Bi atoms remain on the NW, while the
Bi atoms landing on the Si substrate tend to get directly desorbed
again from the surface.

We conclude that the crystal-phase selective Bi incorporation
on GaAs NWs is a result of the interplay between the atomic
morphology and the surface energy of the different NW facets. In
our model, the self-selected diffusion paths and incorporation
sites for Bi atoms are determined by the surface energy and the
next nearest neighbor geometry. By carefully tailoring Wz/Zb
heterostructure NWs, unique 2D and 1D nanostructures of pure
GaBi which are fully confined to one atomic layer can be
achieved, growing self-selectively on the NW {11$\bar{2}$0} facets. A
high density of step edges facing [0001]A/B directions is found
essential to trigger the controlled Bi incorporation. The crystal
phase and surface step engineering of NW facets are inspiring for
realization of well-positioned nanostructures and QDs down to
atomic scale. What's more, the study indicates a promising
approach to realize stable and ordered GaBi surface alloys as well
as 3D GaBi nanostructures with atomic-scale precision by using
tailored NW-based templates and radial overgrowth, which is an
important step towards Bi-based III–V nanostructures with novel
topologic and electronic properties.

## Methods
**NW growth**. GaAs NWs were prepared by metal-organic vapor phase epitaxy
(MOVPE) using a 3 × 2″ Aixtron close-coupled showerhead reactor (CCS) at a
total reactor flow of 8 slm, a total reactor pressure of 100 mbar, and hydrogen as
carrier gas. Au aerosol particles with a total areal density of 1 μm$^{-2}$ and nominal
diameters of 30, 50, and 70 nm were deposited onto GaAs($\bar{1}$1$\bar{1}$) substrates by
aerosol technique[30] in order to enable the particle-assisted NW growth mode. After

annealing the substrates at a set temperature of 630 °C for 10 min in an $AsH_3/H_2$ atmosphere, a step necessary to remove surface oxides and allow for a proper surface conditioning, the temperature was set to 550 °C for NW growth. The precursor materials trimethylgallium (TMGa) and arsine ($AsH_3$) were set to molar fractions of $\chi_{TMGa} = 1.9 \times 10^{-5}$ as well as $\chi_{AsH3} = 4.5 \times 10^{-5}$ and $4.4 \times 10^{-3}$ for Wz and Zb conditions, respectively, and, after a temperature stabilization step, introduced into the reactor chamber in order to initialize NW growth. Heterostructured NWs consisting of alternating segments of Wz and Zb, specifically designed for the experiments, were engineered by only switching the group-V precursor flow. For detailed information about the growth of sharp crystal structure interfaces in III–V NW systems, the reader is kindly referred to ref. [38] and references therein. In the presented work the focus was on two types of heterostructured GaAs NWs. Type I had Wz facets of a length of around 43 nm with many small terraces on the NW facets of typically 2–8 nm resulting in a high density of atomic surface steps. Type II sample instead had WZ facets of a length of around 35 nm with terraces typically larger than 8 nm, corresponding to significantly fewer atomic surface steps. Here the terrace size refers to the length of a surface terrace along the direction perpendicular to the NW growth direction. Scanning electron microscopy (SEM) for morphological and structural characterization was carried out in standard and transmission imaging mode in dedicated microscopes (Zeiss Leo Gemini 1560 and Hitachi SU 8010).

**Scanning tunneling microscopy.** For STM imaging, the NWs were broken off their growth substrates, mechanically transferred onto n-type GaAs ($\bar{1}\bar{1}\bar{1}$) substrates[14], and loaded into the STM chamber with a base pressure of less than $1 \times 10^{-10}$ mbar. After this, the samples were annealed to 500 °C for 1–3 h in an atomic hydrogen beam, provided by a thermal cracker (MBE Komponenten) operating at a hydrogen chamber pressure of $2 \times 10^{-6}$ mbar, in order to remove the native oxide which forms on the NW surfaces upon transport in air. This procedure has been proven before to be a suitable way to clean III–V NWs[14,16,18]. The cleaned NW surfaces were studied using an Omicron low-temperature (LT) STM, operated in constant current mode at 5 K. Chosen values of sample bias, $V_T$, and tunneling current, $I_T$, are indicated in the figure captions. Tungsten tips were employed that had been electrochemically etched and afterward cleaned in UHV by electron bombardment. Significant tip-induced band bending effects were observed; due to dopant freeze-out, relatively high sample bias was required upon STM imaging. The Bi deposition was performed on the oxide-free sample, making use of a multi-chamber UHV cluster tool without breaking vacuum, using an effusion cell with a PBN crucible at a sample temperature of 250 °C. The temperature of the Bi cell was set to 420 °C (450 °C for the sample with the larger surface terraces), resulting in an atomic Bi-flux of 4.0 (7.8) $\times 10^{13}$ atoms $cm^{-2} min^{-1}$. The deposition time was 363 s (187 s), corresponding to a deposited Bi amount of 0.5 ML (for both samples).

## Data availability

The STM and XPS raw data generated in this study are available from the corresponding author upon reasonable request.

## Code availability

No custom code was developed for the analysis of the data of this study.

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

## Acknowledgements

This work was performed within the NanoLund Center for Nanoscience at Lund University and was further supported by the Swedish Research Council (VR), grant nos. 2014-04580 and 2017-04108, by the Knut and Alice Wallenberg Foundation (KAW), Grant no. 2017.0061, and by the project CALIPSOplus from the EU Framework Programme HORIZON 2020, Grant Agreement 730872. The STM studies at UCSB were supported by National Science Foundation (NSF) Grant No. DMR-1507875. The authors are grateful to Matteo Amati, Rahul Parmar, and Luca Gregoratti from the Elettra Synchrotron for experimental support.

## Author contributions

Y.L., J.V.K., A.M., and R.T. conceived the idea and developed the layout of the experiment. S.L. grew the NW samples, Y.L., J.V.K., N.W., and E.Y. performed the Bi deposition process and operated the STM system. Y.L. and J.V.K. analyzed the data., R.T., A.M., and C.J.P joined the discussions. Y.L. and J.V.K. prepared the manuscript in collaboration with R.T. All authors discussed the results and commented on the manuscript.

## Funding

## Competing interests

The authors declare no competing interests.
