## [Peer Review File · Nature Communications]

REVIEWER COMMENTS

Reviewer #1 (Remarks to the Author):

The authors study the incorporation of bismuth atoms on GaAs nanowire facets. The bismuth atoms are incorporated on the facets of the nanowires, replacing an arsenic atom. For the hexagonal segments (the wurtzite crystal structure), this substitution is done mainly at the level of the edges of the steps. They can thus form long Ga-Bi chains along the steps.

These results are interesting even if the figures illustrating them are not always extremely convincing. It is very difficult to recognize the structure of the cubic crystal lattice (structure Zinc Blende) in the diagram 2B. It is really necessary to make it more clear and understandable. I do not recognize the 002 direction carrying the GaAs doublet when the structure is observed in zone axis $\langle 110 \rangle$. It is very difficult to recognize on the experimental image Figure 2a, the diagram presented in Figure 2b showing the attachment of bismuth atom at the edge of the step. I do not understand if the segment in red corresponds to an additional atomic plane and where is it on the experimental image? Same remark for the figure S6, the contrast of the experimental image is completely saturated. It is impossible to realize if the bismuth atoms are fixed mainly on the edge of the step of the terrace. So I recommend greatly improving the quality of figures like diagrams.

Reviewer #2 (Remarks to the Author):

This is a very nice study of Bi incorporation on various facets of GaAs nanowires using meticulous imaging by STM.

The authors presented a qualitative model to explain which facets Bi prefers to be incorporated from a surface energetic viewpoint by comparing wurtzite and zincblende surfaces. My main criticism here is that the number of experiments done here is extremely limited - ie only one deposition of Bi was carried out. The arguments presented would be more convincing if a few more parameter space could also be investigated, such as more incorporation of Bi and deposition at different temperatures. Changing the deposition conditions would affect the surface energetics.

The authors associated the 40 - 50 pm protrusion of the Bi atoms as proof that Bi-As exchange has taken place and GaBi is formed. Is there another more direct/convincing way to prove this is actually the case?

Lines 157-158: "Instead adsorption in random sites on the terrace becomes relatively more likely, as well as Bi-Bi clustering, where Bi atoms are incorporated outside group-V lattice sites."

What do the authors mean by Bi-Bi clustering? Are these Bi not having exchange with As? I can only see clustering in the vicinity of the step edges on {110}-type facets.

Response to reviewer Manuscript NCOMMS-21-04306:

Thanks for all reviewers' comments, we are very grateful for the time and effort that all reviewers used on our manuscript and for the constructive critics given. Here are our reply and improvements:

Reviewer #1:

1. "The authors study the incorporation of bismuth atoms on GaAs nanowire facets. The bismuth atoms are incorporated on the facets of the nanowires, replacing an arsenic atom. For the hexagonal segments (the wurtzite crystal structure), this substitution is done mainly at the level of the edges of the steps. They can thus form long Ga-Bi chains along the steps. These results are interesting even if the figures illustrating them are not always extremely convincing. It is very difficult to recognize the structure of the cubic crystal lattice (structure Zinc Blende) in the diagram 2B. It is really necessary to make it more clear and understandable. I do not recognize the 002 direction carrying the GaAs doublet when the structure is observed in zone axis $\langle 110 \rangle$."

Answer:

We strongly appreciate that the reviewer considers our results as interesting.

We agree that it is difficult to recognize the unit cell of the cubic zincblende structure in the 2D cut displaying the (110) plane, as shown in Figure 2B (now 2C). A similar argument holds for the (11-20) plane of the wurtzite structure shown in Figure 2D (now 2F). Therefore, we added ball-and-stick 3D models of the zincblende (new Figure 2B) and the wurtzite (new Figure 2E) crystal structure as shown below, seen from the [110] and [11-20] direction, respectively, to better illustrate how the surfaces relevant for this study belong to the corresponding zincblende and wurtzite crystal structures.

The crystal facets of the nanowire look like this:

In both figures, the atomic ball-and-stick model is in the lower part overlaid by the nanowire shape (not to scale). In 2B, the shape is determined by {110} site facets with a 60° angle between adjacent facets. The $\langle 111 \rangle$ B growth direction is indicated in the image. The cubic unit cell of the zincblende structure is indicated by black lines. To show the unit cell more clearly, the two most outside

(towards the reader) and two most inside As atoms are not shown. In 2E, both {11-20} (blue) and {10-10}(grey) site facets of the nanowire shape are shown, with a 30° angle between adjacent facets. The hexagonal unit cell is indicated by black lines.

The figure caption of Figure 2 has been updated as follows:

"Figure 2. Bi incorporates into the surface layer of GaAs NWs. (A) and (D) show STM images of the Zb {110}-type surface and of the {112⁻0}-type surface, respectively, including surface steps along different directions between the top terrace and a one monolayer lower terrace. The lower terraces appear black here due to contrast enhancement, the height scale bars are shown on the right side. Different contrast settings of the same STM image can be found for comparison in Fig. S6 in the SI. The many small bright protrusions correspond to Bi atoms. Red arrows in (A) and (D) point at atomic vacancies. The inset in (D) indicates a 2D GaBi island near a vacancy in the middle of a flat terrace. (B) and (E) illustrate the GaAs 3D crystal structure for (B) Zb and (E) Wz phase, overlaid with the NW shape. (C) and (F) show 2D models corresponding to the surfaces seen in (A) and (D). The red and blue areas represent the top terrace and the second atomic layer, respectively, of the NW surface, separated by surface steps. An arbitrary shape is chosen for the top terrace, but the possible directions of the surface steps are determined by the crystal structure and correspond to those visible in (A) and (D). Incorporation of Bi atoms (yellow dots) is illustrated, occurring at vacancies and via atomic step edges facing <111>A/B directions on the Zb{110}-type surface (C) and step edges facing the [0001] B growth direction on the Wz {112⁻0}-type surface (F), as motivated by the positions of the Bi atoms observed in (A) and (D). The direction indexes with red underline in (C) present the projections of <111>-type directions in {110} plane. In (B), (C), (E), and (F), pink (green/yellow) spheres depict Ga (As/Bi) atoms. UT = -4.4 V, IT = 100 pA for all STM images."

Furthermore, the following text has been added to the manuscript:

"The atomic lines and zig-zag pattern in Fig. 2A and Fig. 2D are due to the specific side facets exposed in NWs, the corresponding crystal structure models can be seen in Figs. 2B and 2E, respectively."

Also the 2D model of Fig. 3C has been replaced by a 3D ball-and-stick model of the atomic structure overlaid with the nanowire shape.

2. *"It is very difficult to recognize on the experimental image Figure 2a, the diagram presented in Figure 2b showing the attachment of bismuth atom at the edge of the step. I do not understand if the segment in red corresponds to an additional atomic plane and where is it on the experimental image?"*

Answer:

Yes, the red area corresponds to another monolayer of atoms on the top of the blue flat layer. The model in Fig. 2B (Fig. 2C in the revised version) just shows an ideal case which has step edges in many different directions for illustration. It does not refer to an identical atomic terrace shape in the STM images, but the possible directions of the step edges correspond to those found experimentally for

the different surface structures. We have modified the caption to make this more clear, please see the end of the previous answer.

3. *"(follow the last comment)Same remark for the figure S6, the contrast of the experimental image is completely saturated. It is impossible to realize if the bismuth atoms are fixed mainly on the edge of the step of the terrace. So I recommend greatly improving the quality of figures like diagrams."*

Answer:

We agree that the position of the Bi atoms of the top terrace cannot be seen in this figure, instead the contrast is enhanced for the atoms of the lower terrace. We forgot to mention that the figure shows the same STM image as Fig. 2(D) of the main manuscript, this information has now been included. Furthermore, we added a third illustration of the same STM image as Fig. S6(B), this one with reduced contrast, which allows all atomic structures of both terraces to be recognized. The height scale bar is shown on the right side in image B, the dark and bright yellow colors correspond to the top atomic terrace consisting of GaAs and GaBi, respectively, while the lower terrace has dark and bright blue color. Some incorporated Bi sites are indicated with red arrows, while the yellow arrows point out the shadow of the step edges due to double tip effect. The figure caption has been

updated as follows:

"Figure S6. (A) An STM image of a Wz {11 $\bar{2}$ 0}-type facet is shown in the image, where positions of all Bi atoms are indicated and marked with counting numbers. This STM image contributes to the statistics of Bi sites density as "nanowire 3- Wz {11 $\bar{2}$ 0} facet" in Table S2. The figure shows the same STM image as in Fig. 2(D) of the main manuscript, but with different contrast settings, to highlight Bi atoms in the lower atomic terrace. (B) The same STM image as in (A), but with reduced contrast, in order to visualize the atomic structures of both terraces. A height scale bar is shown. Some incorporated Bi sites are indicated with red arrows, and the yellow arrows point out some shadows of the step edges due to double tip effect."

The existence of the alternative illustration with different contrast settings is now also mentioned in the figure caption for Fig. 2(D)(see above).

Reviewer #2:

1. *"This is a very nice study of Bi incorporation on various facets of GaAs nanowires using meticulous imaging by STM. The authors presented a qualitative model to explain which facets Bi prefers to be incorporated from a surface energetic viewpoint by comparing wurtzite and zinblende surfaces."*

Answer: We highly appreciate the acknowledgement of our experiments.

2. *"My main criticism here is that the number of experiments done here is extremely limited - ie only one deposition of Bi was carried out. The arguments presented would be more convincing if a few more parameter space could also be investigated, such as more incorporation of Bi and deposition at different temperatures. Changing the deposition conditions would affect the surface energetics."*

Answer: We agree with the reviewer that a closer and systematic exploration of the parameter space for Bi deposition is desirable. However, we have to point out that the complex and time-consuming nature of our STM experiments on nanowires, including the need to navigate with the STM tip to a nanowire and find the appropriate crystal phase segment of this nanowire, does not allow a strongly systematic approach or higher statistics. Still, we have explored the relevance of some of the growth and deposition temperatures, such as sample temperature, sample topography, and deposition rate. We found that if the sample is at too low temperature (e.g. room temperature) during Bi deposition, only metallic Bi clusters will be formed on the surface. If the temperature is too high (like 400°C), Bi atoms tend to evaporate away directly when landing on the GaAs surface, thus resulting a very low Bi deposition and incorporation rate. In the end, we aimed 250°C as a propriate sample temperature during incorporation, since it insures a decent Bi deposition rate and offer high enough surface energy to trigger the Bi incorporation reaction. The temperature of the Bi source has been tested as well, where we found that a source temperature of 400°C gave good control of the Bi deposition rate for our experimental setup with a distance of about 15-20 cm between the top of Bi PBN cell and the sample: around half a monolayer of Bi could be achieved in a few minutes. The following text has been added to the manuscript, as a last paragraph before the conclusion:

Finally, we want to further explore the conditions under which Bi incorporation into the sample surface can occur at all. We found that if the sample is at too low temperature (e.g. room temperature) during Bi deposition, only metallic Bi clusters will be formed on the surface. On the contrary, if the sample temperature is too high (e.g. 400°C), Bi atoms tend to evaporate directly when landing on the GaAs surface, thus resulting in a very low Bi deposition and incorporation rate. Synchrotron based nano-focused X-ray photoelectron spectroscopy (XPS)..." [see continuation in the reply to point 3]

In the manuscript, we focused on the influence of the atomic structure and larger topography of the sample surface as a very important part of the parameter space for Bi incorporation. Among all the experiments we have done in different conditions, we choose two typical and successful experimental results to be shown in the manuscript: Bi incorporation on the GaAs NW surface with smaller terrace size (higher step density), and larger terrace size (lower step density). These

two patches of NWs were tailored grown especially for our study. In the end, the clear differences (Fig.2 and Fig.3 in manuscript) demonstrate well how the step edges effect the Bi incorporation process, supporting our main result in the study of the Bi incorporation mechanism.

In the meantime, we have obtained more experimental results under different Bi deposition conditions, which are shown below and which also are added to the Suppl. Inf. Of the manuscript (section 9) as Fig. S8. In figure A and B, STM images of Wz{11-20}-type nanowire facets are shown, which were exposed to a shorter Bi deposition time, resulting in only about 10% coverage (in theory). The brightness range has been adjusted so it shows the top two terraces of GaAs: the yellow layer is sitting on the top of the blue layer. A height profile is taken in figure A and B and shown in figure C and D, respectively. Some of the Bi sites are pointed out with red arrows, both in the STM images and in the height profile curve.

In figure A, GaBi structures are formed on the step edges facing $[0001]A/B$ directions, similar to those seen in Fig. 2D of the main manuscript, which shows the initial state of Bi incorporation. Due to the much lower Bi coverage, the structure size is by far too small to be called a 2D GaBi nanostructure. In figure B, there are some small GaBi structures containing 3-5 Bi atomic sites scattered on the surface, however, not connected yet. These results are also in agreement with the Bi incorporation mechanism discussed in the manuscript.

The following text has been added to the main manuscript:

“Further STM measurements, performed on Wz $\{11\bar{2}0\}$ -type NW facets after Bi deposition resulting in a coverage of less than 10% (in theory), can be seen in Fig. S8 of the SI. They confirm that the initial sites for Bi incorporation are step edges facing $[0001]A/B$ directions.”

We also tried a higher Bi coverage, that was done by repeating another Bi deposition after a proper 1st Bi deposition (around half a monolayer in theory). After that, the surface was found to be covered with amorphous clusters, making it difficult to resolve individual Bi sites. This can be due to the surface contamination after the sample was taken out from LT-STM at 10 K, or due to strong Bi-Bi bonds forming metallic Bi clusters.

3. *“The authors associated the 40 - 50 pm protrusion of the Bi atoms as proof that Bi-As exchange has taken place and GaBi is formed. Is there another more direct/convincing way to proof this is actually the case?”*

Answer: Using the height difference in STM image in atomic scale has been a widely used and solid approach on identifying atom species. Similar examples can be seen in other studies of Bi incorporation^{1,2} and Sb incorporation^{3,4}. In our experiments, we not only observe some ~50 pm protrusions in atomic scale, but the high-resolution images also show that the protrusions are exactly located on As sites (not on Ga sites, nor random places between two sites). Generally, the contrast in STM images is determined by electronic contrast (local density of states, LDOS) and surface topography. LDOS varies according to tip-sample bias. In our experiment, we were continuously scanning the same area with different bias, and it turned out that the apparent height of the Bi site protrusion was constant at around 50 pm. Thus, we can exclude the influence from the variation of LDOS, and conclude the protrusion indeed comes from surface structure, i.e. due to the different bond length of Ga-As and Ga-Bi.

Besides, we have performed XPS with an X-ray beam size around 120 nm, providing nanofocus XPS and scanning photoelectron microscopy (SPEM), in order to explore surface chemical changes along the nanowire and compared to its surrounding. These experiments have been done at the Elettra synchrotron in Trieste, Italy, the results and their discussion have been added to the SI as section 10. GaAs nanowires were mechanically transferred from their growth substrate onto Si substrates containing metal markers, so that the same NW could be located in SPEM images before and after Bi deposition. Figure A below (Figure S9A of the SI) shows a scanning electron microscopy image of a specific nanowire together with its structural model, containing both Wz and Zb segments. Figure B shows XPS spectra of the Bi 4f core-level, acquired in the middle of the NW shown in figure A, before and after Bi deposition. We can see a distinct Bi 4f doublet after Bi deposition (the blue spectra), obtained from the NW. The XPS intensity is relatively low because the signal only comes from one NW with about half a monolayer of Bi on the surface. We also performed 2D SPEM mapping of Bi 4f and Ga 3d core-levels on the same NW, which are shown in Fig. C and D, respectively. The color scale corresponds to the distribution of the Bi and Ga signal in 2D. We can see that both images show the contrast in the same shape and size as the nanowire. Figure C indicates that the majority of Bi 4f signal comes from the nanowire and that there is almost no Bi staying on the Si substrate. This agrees with the STM results showing that Bi atoms have incorporated into the NW surface and formed chemical bonds, while most of the Bi atoms that have been evaporated onto the Si substrate simultaneously seem to have desorbed again.

In the STM images in the manuscript, we observe no sign for large scale Bi clustering, the large majority of the Bi atoms visible at the surface has been incorporated on the As sites. Considering the clear XPS Bi signal from the nanowire surface, we attribute it to Bi atoms bonding to Ga, and we draw the conclusion that the 50 pm protrusions in the STM images come from the incorporated Bi atoms and the formed Ga-Bi bonds.

The following text has been added to the main manuscript:

"Synchrotron based nano-focused X-ray photoelectron spectroscopy (XPS) has been performed to study the Bi incorporation process as well, as discussed in section 10 of the SI. The presence of Bi atoms in the NW surface after Bi deposition was confirmed from Bi 4f core level spectra. For these experiments, GaAs NWs have been transferred onto Si substrates prior to Bi deposition. Interestingly, 2D maps obtained by scanning photoelectron microscopy (SPEM), shown in Fig. S9 of the SI, demonstrate a much higher Bi signal at the GaAs NW as compared with the surrounding Si substrate. This can be explained by the Bi incorporation process on NWs, such that the incorporated Bi atoms remain on the NW, while the Bi atoms landing on the Si substrate tend to get directly desorbed again from the surface."

4. Reviewer #2: Lines 157-158 (Instead adsorption in random sites on the terrace becomes relatively more likely, as well as Bi-Bi clustering, where Bi atoms are incorporated outside group-V lattice sites), what do the authors mean by Bi-Bi clustering? Are these Bi not having exchange with As? I can only see clustering in the vicinity of the step edges on {110}-type facets.”

Answer: Yes, we refer to these clustering Bi features in the vicinity of step edges on (110)-type facets. Our interpretation is that these Bi atoms have not undergone Bi-for-As exchange or any other chemical reaction with the GaAs surface but just stay as adsorbents, sitting on the surface. In our experiment we found out that Bi atoms are very mobile and can be unstable on a very hot GaAs surface. For example, if Bi atoms are deposited on substrate in higher temperature (e.g., 400°C), the Bi deposition rate dramatically decreases, even under the same Bi flux. The unreacted Bi atoms which stay in the form of Bi-Bi clusters after deposition at lower temperatures can also desorb again when annealed to 400°C. XPS results from GaAs(110) substrates show that Bi atoms with Bi-Ga bonds are dominant after deposition of smaller amounts of Bi onto GaAs heated to 250°C, while after deposition of more than one monolayer of Bi the Bi-Bi chemical state dominates instead. Further annealing of the sample to 400°C reduces the intensity of the Bi-Bi state again, corresponding to desorption of the Bi-Bi clusters or larger islands. Coming back to our STM images, we explain the phenomenon that Bi cluster only exist on the ZB step edges such that the unreacted Bi atoms keep mobile on the flat surface until they are trapped by other Bi atoms or clusters at the step edges.

Other changes:

1. We update the model image in Fig. 3C for better visualization.
2. Height scale bars are added to Fig. 2A, D, Fig. 3A, B in the main manuscript, as well as Fig. S2, S5 in the SI.
3. The abstract was slightly modified to meet the format restrictions of Nature Communications.
4. We added an acknowledgments section.
5. A data availability statement was added.

References:

- 1 Lin, A., Doty, M. F. & Bryant, G. W. Incorporation of random alloy GaBi(x)As(1-x) barriers in InAs quantum dot molecules: Energy levels and confined hole states. *Physical Review B* **99**, 075308, doi:10.1103/PhysRevB.99.075308 (2019).
- 2 Bastiman, F., Cullis, A. G., David, J. P. R. & Sweeney, S. J. Bi incorporation in GaAs(100)-2×1 and 4×3 reconstructions investigated by RHEED and STM. *Journal of Crystal Growth* **341**, 19-23, doi:10.1016/j.jcrysgro.2011.12.058 (2012).
- 3 Timm, R. *et al.* Quantum ring formation and antimony segregation in GaSb/GaAs nanostructures. **26**, 1492, doi:10.1116/1.2952451 (2008).

- 4 Hjort, M. *et al.* Crystal Structure Induced Preferential Surface Alloying of Sb on Wurtzite/Zinc Blende GaAs Nanowires. *Nano Lett* **17**, 3634-3640, doi:10.1021/acs.nanolett.7b00806 (2017).

REVIEWERS' COMMENTS

Reviewer #1 (Remarks to the Author):

The modifications made to the texts and in particular to the figures are satisfactory and largely answer my questions and requests for clarification. Therefore I recommend to the editorial committee the publication of this paper.

Reviewer #2 (Remarks to the Author):

The authors have done a superb job addressing the reviewers' comments. New experimental results have further supported the discussion/conclusion in the manuscript.

I recommend the manuscript for publication.

Response to reviewer:

Reviewer #1:

The modifications made to the texts and in particular to the figures are satisfactory and largely answer my questions and requests for clarification. Therefore I recommend to the editorial committee the publication of this paper.

Response: We are happy to read that we could answer the reviewer's question and we acknowledge his/her reply and effort viewing our manuscript.

Reviewer #2:

The authors have done a superb job addressing the reviewers' comments. New experimental results have further supported the discussion/conclusion in the manuscript. I recommend the manuscript for publication.

Response: We are grateful to the reviewer for his/her very positive reply and efforts in reviewing our work!